# A deep learning algorithm with good prediction efficacy for cancer-specific survival in osteosarcoma: A retrospective study

**Yang Liu[1]ᵒ, Lang Xie[2]ᵒ, Dingxue Wang[3], Kaide Xia[4]***

**1** Department of Orthopedics, The First Affiliated Hospital of Guizhou University of Traditional Chinese Medicine, Guiyang, China, **2** Hospital Infection Management Department, Bijie First People's Hospital, Bijie, China, **3** Department of Oncology, The First Affiliated Hospital of Guizhou University of Traditional Chinese Medicine, Guiyang, China, **4** Clinical College of Maternal and Child Health Care, Guizhou Medical University, Guiyang, China

ᵒ These authors contributed equally to this work.
* xiakaide@163.com

**Data Availability Statement:** All data are available from the Surveillance, Epidemiology, and End Results (SEER) Program (www.seer.cancer.gov) SEER*Stat Database: Incidence - SEER 18 Regs Custom Data (with additional treatment fields), Any

## Abstract

### Objective

Successful prognosis is crucial for the management and treatment of osteosarcoma (OSC). This study aimed to predict the cancer-specific survival rate in patients with OSC using deep learning algorithms and classical Cox proportional hazard models to provide data to support individualized treatment of patients with OSC.

### Methods

Data on patients diagnosed with OSC from 2004 to 2017 were obtained from the Surveillance, Epidemiology, and End Results database. The study sample was then divided randomly into a training cohort and a validation cohort in the proportion of 7:3. The DeepSurv algorithm and the Cox proportional hazard model were chosen to construct prognostic models for patients with OSC. The prediction efficacy of the model was estimated using the concordance index (C-index), the integrated Brier score (IBS), the root mean square error (RMSE), and the mean absolute error (SME).

### Results

A total of 3218 patients were randomized into training and validation groups (n = 2252 and 966, respectively). Both DeepSurv and Cox models had better efficacy in predicting cancer-specific survival (CSS) in OSC patients (C-index >0.74). In the validation of other metrics, DeepSurv did not have superiority over the Cox model in predicting survival in OSC patients.

registered researcher can free download from https://seer.cancer.gov/data/.

**Funding:** The authors received no specific funding for this work.

**Competing interests:** The authors have declared that no competing interests exist.

## Conclusions

After validation, our CSS prediction model for patients with OSC based on the DeepSurv algorithm demonstrated satisfactory prediction efficacy and provided a convenient webpage calculator.

## 1. Introduction

Osteosarcoma (OSC) is among the most frequently observed primary tumors of the bone in children and adolescents and ranks third after chondrosarcoma and chordoma in adults [1]. The tumor is usually located in the distal femur and proximal tibia, with a survival rate of 50% to 65%, but 25%-50% of patients with initial metastases die from pulmonary metastases [2]. OSC shows a bimodal age distribution, with the first peak at 15–19 years and the second at 75–79 years [3, 4]. OSCs are derived from primitive mesenchyma cells, often in bones and rarely in soft tissues [5]. Although local and distant OSC metastases progress slowly, the presence or absence of metastases is an important prognostic factor [6]. The staging system of the American Joint Committee on Cancer (AJCC) is recommended by The World Health Organization; however, it has shortcomings in regards to focus and usefulness for predicting patient prognosis. Several previous studies have used nomograms to predict cancer patient survival and have achieved positive predictive efficacy [7–9]. The nomogram is a Cox proportional hazard model (CPH), and its premise is based on the following: a restrictive assumption of proportional hazard between the independent and dependent variables is satisfied. However, it is difficult to identify the practical fundamental relationship between the two variables in practice. In addition, a linear relationship between clinical characteristics and prognostic outcomes alone is not sufficient for clinical decision-making [10]. Hence, there is a need for a better model that evaluates the relationship between these nonlinear variables.

The deep learning network provides new perspectives on how to address the highly complicated linear or nonlinear relationships between clinical features and prognostic hazards of individuals [11, 12]. Fotso et al. developed a Python-based deep neural network called PySurvival [13], and it is useful for predicting the impact of patient characteristics on prognosis. In addition, the authors confirmed that the algorithm showed better performance than other methods in handling survival data. Currently, this algorithm has performed well in several cancer prognostic studies [14, 15].

To date, we have found no reports on the application of DeepSurv to OSC prognosis. Therefore, the objective of this research was to develop a DeepSurv-based prognostic model for cancer-specific survival (CSS) for patients with OSC using patient data from the Surveillance, Epidemiology, and End Results (SEER) database, and to compare the efficacy of DeepSurv with that of the Cox proportional hazard model to provide physicians and patients with predictive tools to assess the risk stratification and individual prognosis of patients with OSC.

## 2. Materials and methods

### 2.1. Eligibility criteria and clinical information

Study data were extracted from the SEER database, Plus version (https://seer.cancer.gov/), in which 18 states are enrolled, and released in April 2022 [15]. Criteria such as the primary tumor site and histological information were selected according to the International Classification of Tumor Diseases, 3rd ed. (ICD-O-3). Criteria for inclusion were defined as the following:

(1) the primary site was coded as C40-41, (2) the histological codes were 9180–9187 and 9192–9194, (3) the year of diagnosis was between 2004 and 2017, and (4) the behavioral code was malignant. Criteria for exclusion were listed as: (1) missing data on months of survival, race, surgery status, or year of diagnosis; (2) imprecise tumor size; (3) unclear T-, N-, or M-stage status; or (4) laterality status listed as "missing laterality information," or "bilateral tumors."

Because the SEER database withholds private patient information and the authors have signed an official data-use agreement with the database, no further ethical review by the authors' institutions was required.

## 2.2. Selection and reconfiguration of variables

Fifteen characteristic variables were included: age at diagnosis, sex, race, marital status, number of tumors, T stage, N stage, M stage, grade, SEER combined stage, primary site, radiation, surgery, and chemotherapy. The characteristic variables covered the demographic, clinical data, and treatment information of patients with OSC. The continuous variables of age and tumor size were classified using the X-tile software (https://medicine.yale.edu/lab/rimm/research/software/) to determine the best cut-off values. Marital status was classified as "married" or "other," and tumor number was classified as "single" or "multiple." The primary site was classified according to the actual frequency distribution; tumors with a higher frequency were classified separately and those with a frequency <150 were combined as "other." Surgical modalities were classified into three categories: non-operated, radical surgery, and other surgery. Cancer-specific survival was the endpoint of interest, defined as the time interval between diagnosis and death due to OSC.

## 2.3. Model development and performance evaluation

In this study, two algorithms were selected for training: DeepSurv and Cox proportional hazard (CPH). The dataset was randomly split into training and validation sets in a ratio of 7:3. The predictive performances of the models were evaluated using the concordance index (C-index), the integrated Brier score (IBS), the root-mean-square error (RMSE), and the mean absolute error (MAE). The C-index varies between 0 and 1, and the closer to 1, the better the discriminatory ability of the model. The IBS is calculated by integrating the prediction-error curve between 0 and 0.25, with values closer to 0 indicating more precise prediction performance of the model [16]. The RMSE and MAE describe the differences between the actual and predicted CSS values in each model, with smaller values indicating better model performance [17].

## 2.4. Statistical analysis

Categorical variables are summarized as frequencies (n) and percentages (%). Differences between the two groups in baseline characteristics were analyzed using the chi-square test. Data cleaning and time-dependent ROC curves were generated using R software (version 4.2; https://www.r-project.org/). Survival algorithms were implemented in Python (version 3.7; https://www.python.org/) using PySurvival. Statistical significance was set at $p < 0.05$.

# 3. Results

## 3.1. Patient characteristics

A total of 3614 patients with OSC were identified in the SEER database as eligible for inclusion. After application of the exclusion criteria, 3218 of these were analyzed. Random splitting in a 7:3 ratio produced 2252 patients in the training cohort and 966 patients in the validation cohort (**Fig 1**).

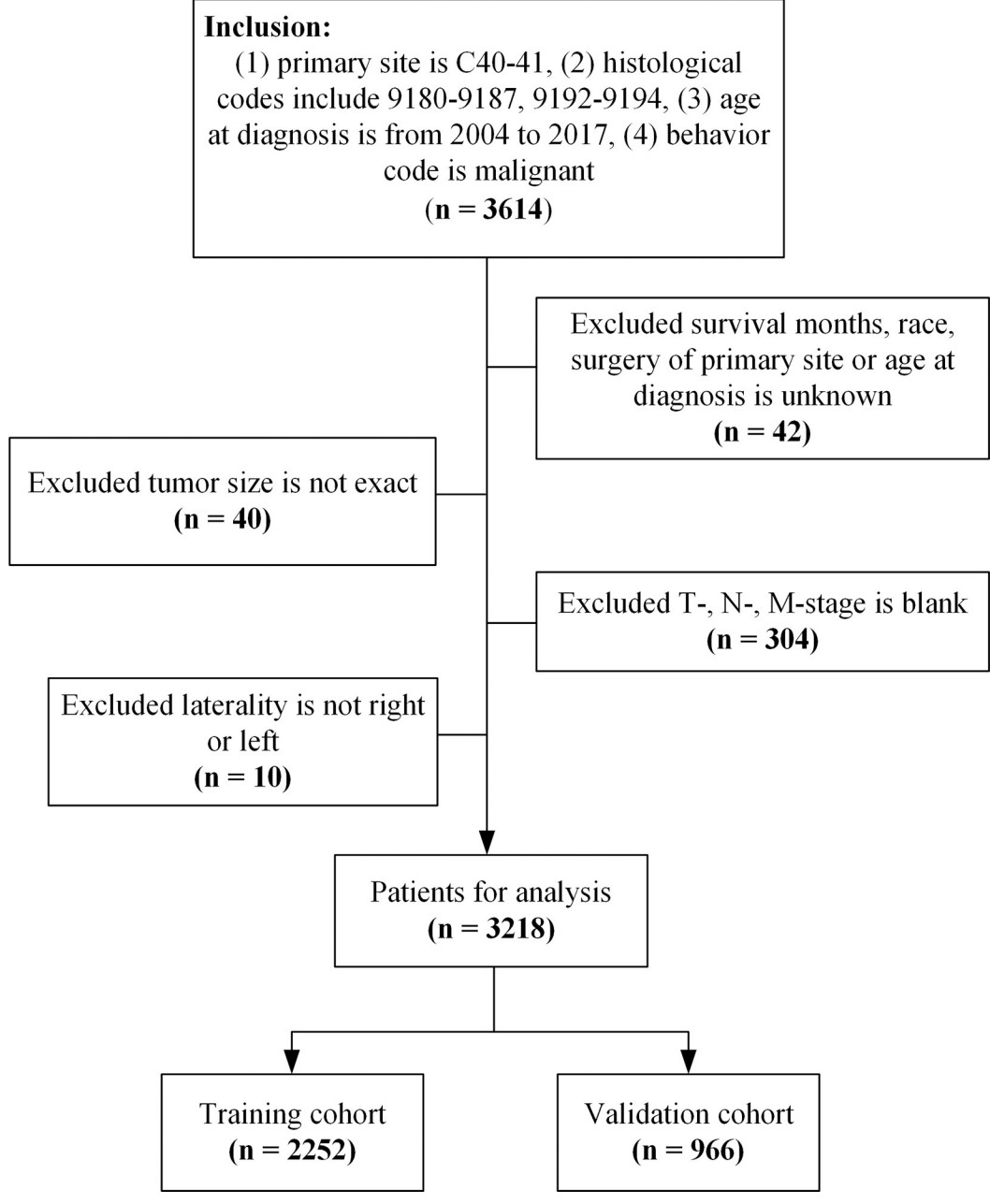

**Fig 1. Patient selection.**

The characteristics of the final sample set were 68.2% <35 years of age, 54.7% men, 60.8% with tumors less than 11.8 cm, 59.7% with cancer in the long bones of the lower limb and associated joint sites, 85% with a single primary tumor, 82% with stage T1-2, 81% with distant metastases, and 67.1% with grade III-IV. The N0 stage was observed in 95.9% of patients, an absence of distant metastasis in 81%, grade III-IV in 67.1%, no radiotherapy in 90.6%, and chemotherapy in 80.1%. After randomized splitting, we found no statistical differences in the characteristics of the training and validation cohorts except in the N stage, indicating good comparability between the two cohorts (**Table 1**).

**Table 1. Clinical and pathological characteristics of the study sample of patients with osteosarcoma.**

| Characteristic | Overall (n = 3,218) | Training (n = 2,252) | Validation (n = 966) | P-value |
|---|---|---|---|---|
| Age at diagnosis (y) | | | | 0.930 |
| 35–65 | 713 (22.2%) | 496 (22%) | 217 (22.5%) | |
| <35 | 2,194 (68.2%) | 1,536 (68.2%) | 658 (68.1%) | |
| >65 | 311 (9.7%) | 220 (9.8%) | 91 (9.4%) | |
| Sex | | | | 0.897 |
| Female | 1,458 (45.3%) | 1,022 (45.4%) | 436 (45.1%) | |
| Male | 1,760 (54.7%) | 1,230 (54.6%) | 530 (54.9%) | |
| Race | | | | 0.605 |
| Black | 492 (15.3%) | 353 (15.7%) | 139 (14.4%) | |
| Other | 294 (9.1%) | 202 (9.0%) | 92 (9.5%) | |
| White | 2,432 (75.6%) | 1,697 (75.4%) | 735 (76.1%) | |
| Marital | | | | 0.323 |
| Married | 766 (23.8%) | 547 (24.3%) | 219 (22.7%) | |
| Other | 2,452 (76.2%) | 1,705 (75.7%) | 747 (77.3%) | |
| Tumor size (cm) | | | | 0.443 |
| <11.8 | 1,937 (60.2%) | 1,342 (59.6%) | 595 (61.6%) | |
| ≥11.8 | 799 (24.8%) | 562 (25.0%) | 237 (24.5%) | |
| Unknown | 482 (15%) | 348 (15.5%) | 134 (13.9%) | |
| Primary site | | | | 0.801 |
| C40.0 | 363 (11.3%) | 263 (11.7%) | 100 (10.4%) | |
| C40.2 | 1,921 (59.7%) | 1,330 (59.1%) | 591 (61.2%) | |
| C41.0 | 170 (5.3%) | 120 (5.3%) | 50 (5.2%) | |
| C41.1 | 169 (5.3%) | 118 (5.2%) | 51 (5.3%) | |
| C41.4 | 307 (9.5%) | 213 (9.5%) | 94 (9.7%) | |
| Other | 288 (8.9%) | 208 (9.2%) | 80 (8.3%) | |
| Tumor number | | | | 0.656 |
| Multiple | 470 (14.6%) | 333 (14.8%) | 137 (14.2%) | |
| Single | 2,748 (85.4%) | 1,919 (85.2%) | 829 (85.8%) | |
| T stage | | | | 0.53 |
| T1 | 1,187 (36.9%) | 820 (36.4%) | 367 (38.0%) | |
| T2 | 1,471 (45.7%) | 1,026 (45.6%) | 445 (46.1%) | |
| T3 | 103 (3.2%) | 74 (3.3%) | 29 (3.0%) | |
| TX | 457 (14.2%) | 332 (14.7%) | 125 (12.9%) | |
| N stage | | | | 0.004 |
| N0 | 3,085 (95.9%) | 2,142 (95.1%) | 943 (97.6%) | |
| N1 | 89 (2.8%) | 75 (3.3%) | 14 (1.4%) | |
| NX | 44 (1.4%) | 35 (1.6%) | 9 (0.9%) | |
| M stage | | | | 0.589 |
| M0 | 2,621 (81.4%) | 1,822 (80.9%) | 799 (82.7%) | |
| M1 | 31 (1.0%) | 24 (1.1%) | 7 (0.7%) | |
| M1a | 327 (10.2%) | 235 (10.4%) | 92 (9.5%) | |
| M1b | 239 (7.4%) | 171 (7.6%) | 68 (7%) | |
| Grade | | | | 0.767 |
| I-II | 304 (9.4%) | 211 (9.4%) | 93 (9.6%) | |
| III-IV | 2,158 (67.1%) | 1,504 (66.8%) | 654 (67.7%) | |
| Unknown | 756 (23.5%) | 537 (23.8%) | 219 (22.7%) | |
| SEER stage | | | | 0.801 |

*(Continued)*

**Table 1.** (Continued)

| Characteristic | Overall (n = 3,218) | Training (n = 2,252) | Validation (n = 966) | P-value |
|---|---|---|---|---|
| Distant | 704 (21.9%) | 508 (22.6%) | 196 (20.3%) | |
| Localized | 1,167 (36.3%) | 805 (35.7%) | 362 (37.5%) | |
| Regional | 1,298 (40.3%) | 904 (40.1%) | 394 (40.8%) | |
| Unknown/unstaged | 49 (1.5%) | 35 (1.6%) | 14 (1.4%) | |
| Radiation | | | | 0.139 |
| None/Unknown | 2,914 (90.6%) | 2,028 (90.1%) | 886 (91.7%) | |
| Yes | 304 (9.4%) | 224 (9.9%) | 80 (8.3%) | |
| Surgery | | | | 0.4 |
| No | 485 (15.1%) | 352 (15.6%) | 133 (13.8%) | |
| Other surgery | 1,068 (33.2%) | 742 (32.9%) | 326 (33.7%) | |
| Radical | 1,665 (51.7%) | 1,158 (51.4%) | 507 (52.5%) | |
| Chemotherapy | | | | 0.252 |
| None/Unknown | 640 (19.9%) | 436 (19.4%) | 204 (21.1%) | |
| Yes | 2,578 (80.1%) | 1,816 (80.6%) | 762 (78.9%) | |

*Codes*: C40.0: long bones of the upper limb, scapula, and associated joints; C40.2: long bones of the lower limb and associated joints; C41.0: bones of the skull, face, and associated joints; C41.1: Mandible; C41.4: pelvic bones, sacrum, coccyx, and associated joints.

## 3.2. Model development and validation

To ensure comparability, we incorporated all "dummy" features into the construction of the DeepSurv and CPH models. For the construction of the DeepSurv model, we used xav_uniform as the initial approach and used an adaptation of the moment estimation estimator with a learning rate of 0.00063 for neural network training.

In the training cohort, the C-index of the DeepSurv model exceeded that of the CPH model (0.790 vs. 0.750, respectively). Additionally, the C-index of DeepSurv exceeded that of CPH in the validation cohort (0.747 vs. 0.744, respectively). The IBS of the DeepSurv model was lower than that of CPH in the training cohort (0.14 vs. 0.15, respectively), but was 0.16 for both algorithms in the validation cohort. The RMSE and MAE of the prediction error of DeepSurv were both larger than those of the CPH model. However, the RMSE and MAE of the survival values of the DeepSurv model were 15.367 and 12.569 respectively, which were smaller than those of the CPH model (17.228 and 14.900, respectively) (**Fig 2**). Although the time-dependent ROC area of the DeepSurv model was larger than that of the CPH model in the training set, the event-dependent ROCs of both models overlapped well in the training cohort (**Fig 3**).

## 3.3. Algorithm deployment

Based on the DeepSurv algorithm, we built an application that predicted the CSS of patients with OSC after the entry of relevant information regarding the patient's condition. In addition, the application can easily display the CSS rates of patients at 3, 5, and 10 years. The functionality of the application and visualization of the output are shown in **Fig 4**. This application is primarily intended for purposes of research and information and can be accessed publicly at the following link https://rrreert-1-14-main-1xfl0e.streamlit.app/.

## 4. Discussion

Accurate prediction of the survival of patients with OSC is crucial for counseling, follow-up, and management of treatment. With the development and refinement of machine learning

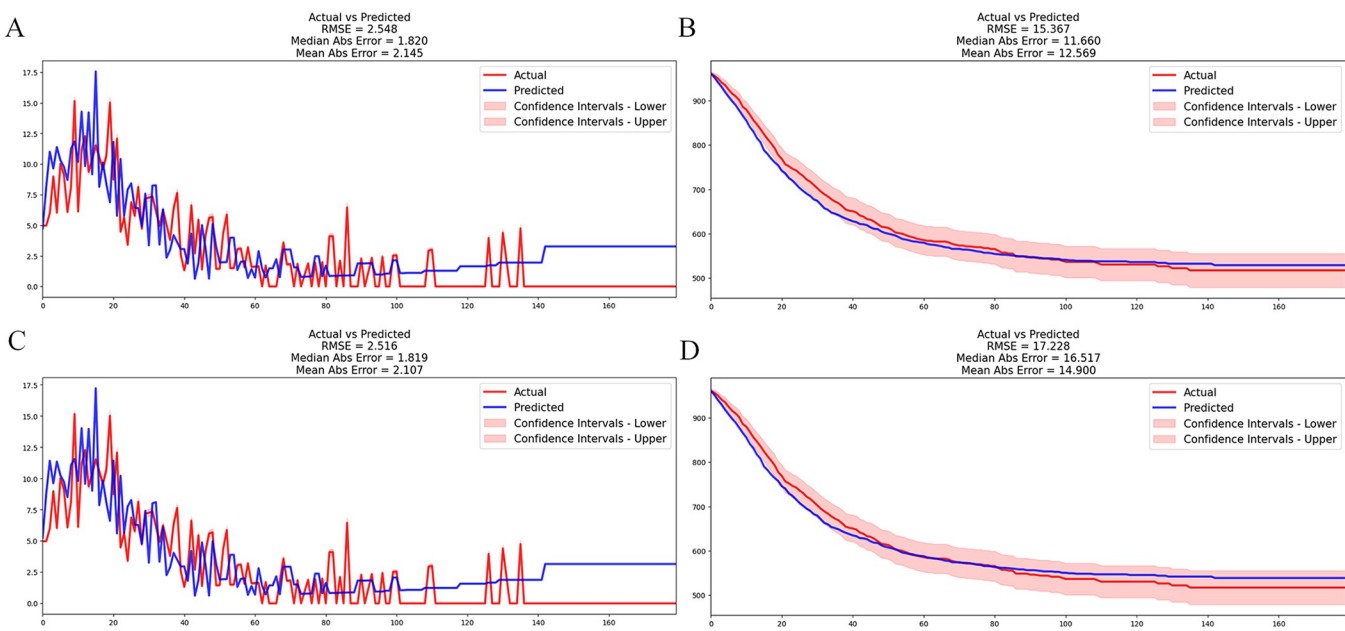

**Fig 2. Performance of the DeepSurv and cox proportional hazard (CPH) models in the validation cohort.** (**A, B**) Prediction errors in the DeepSurv model by root-mean-square error (RMSE) and mean absolute error (MAE), respectively. (**C, D**) Prediction errors in the CPH model by RMSE and MAE, respectively.

algorithms, their applications in the medical field have become increasingly widespread [11, 18, 19]. Due to the use of an increased number of data dimensions and volume of data, machine learning has begun to rival the predictive performance of the traditional CPH model. In the present study, we employed the DeepSurv algorithm to build and evaluate a prognostic model of the CSS rate in patients with OSC, compared its predictive efficacy with that of the CPH model, and demonstrated relatively good predictive efficacy.

Studies have confirmed that age, surgical approach, tumor size, grade classification, primary site, distant metastases, and adjuvant radiotherapy are prognostic factors in patients with OSC [6, 7, 20]. Most of these studies used CPH regression algorithms for prediction, which means

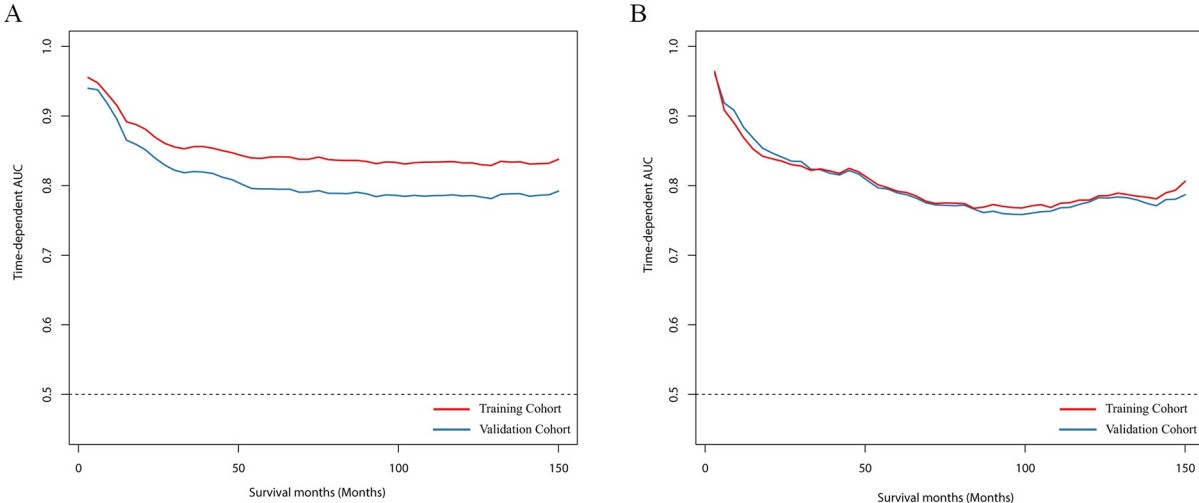

**Fig 3.** The time-dependent ROC curves for (**A**) the training cohorts and (**B**) the validation cohorts.

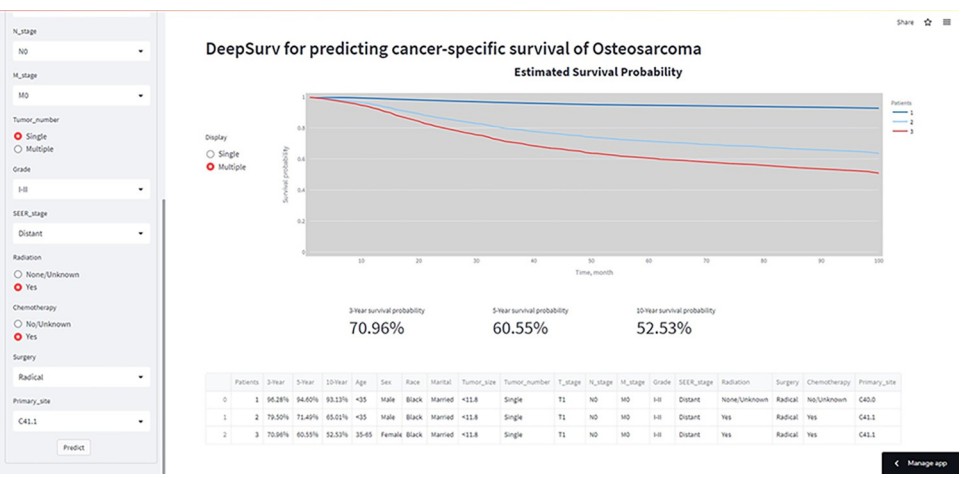

**Fig 4. Interface display of the web-based calculator.**

that the following two effects may have been simplified or ignored: effects correction, the causal effect of one exposure within the levels of another interest exposure; and cross-interaction, the causality of two exposure effects within a domain of interest [21]. Therefore, we used the DeepSurv algorithm to accommodate nonlinearities, reduce interactions, and reduce effect corrections in the SEER queue [22]. The calculator we deployed on a web page allowed not only the prediction of individual CSS rates in patients with OSC but also the comparison of the prognostic impact of different levels and variables. In the present study, the DeepSurv algorithm was not found to be superior to CPH in predicting CSS by the various metrics we evaluated.

In previous studies, machine learning algorithms representative of DeepSurv have outperformed the traditional Cox proportional hazard model in survival prediction [10, 23, 24]. In the training cohort of the present study, the DeepSurv model had a higher C-index than the CPH model; however, in the validation cohort, it did not show improved efficacy in predicting the CSS rates of patients. This suggests that machine-learning algorithms can only show advantages under conditions where traditional models are limited. Several explanations are possible for the similar efficacies observed in DeepSurv and CPH in the present study. First, the number of features used to build the model may not have been sufficiently large enough to demonstrate the advantages of machine learning in dealing with large samples of multidimensional data. Second, the collection of features available from the SEER database was mostly based on clinical experience, suggesting that the features collected may have had a strong linear relationship with patient outcomes. These features may be more suitable for applications using parametric models such as CPH. In testing the model hypothesis, the DeepSurv model was applied under a wider range of conditions than CPH, but achieved a similar predictive efficacy. This implies that DeepSurv may be an effective alternative model for predicting the CSS rate in patients with OSC.

Although we aimed to use the DeepSurv algorithm to predict the survival of patients with OSC, we obtained a model with good performance and subsequently deployed it on a webpage for easy access. However, our study has several limitations. First, it is a retrospective study with potential selection bias. Second, model training and validation were both performed using the SEER database, without external validation. Finally, the dummy-variable form used for fitting the models increased the number of features, resulting in a lack of information about feature importance in the output of the study model. Therefore, there is a significant need to implement a multicenter, large-scale prospective trial to validate the effectiveness of the model.

## 5. Conclusions

Using the DeepSurv algorithm, we developed a high-performance prediction model for CSS rates in patients with OSC. In addition, the developed model was deployed on a webpage to provide physicians and patients with an easy-to-use management prediction tool to facilitate personalized treatment. Our study indicates that the DeepSurv algorithm demonstrates high potential for use in applications in both clinical research and practice.

## Acknowledgments

*We would like to thank Editage (www.editage.cn) for English language editing.*

## Author Contributions

**Data curation:** Lang Xie.

**Funding acquisition:** Kaide Xia.

**Methodology:** Lang Xie.

**Resources:** Yang Liu.

**Software:** Yang Liu, Dingxue Wang.

**Supervision:** Kaide Xia.

**Validation:** Dingxue Wang.

**Writing – review & editing:** Kaide Xia.

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
