## [Decision Letter · Decision Letter 0]

5 May 2023

PONE-D-23-11162A deep learning algorithm with good prediction efficacy for cancer-specific survival in osteosarcoma: a retrospective studyPLOS ONE

Dear Dr. Xia,

Thank you for submitting your manuscript to PLOS ONE. After careful consideration, we feel that it has merit but does not fully meet PLOS ONE’s publication criteria as it currently stands. Therefore, we invite you to submit a revised version of the manuscript that addresses the points raised during the review process.

We look forward to receiving your revised manuscript.

Kind regards,

Filomena de Nigris, Ph.D.

Academic Editor

PLOS ONE

Journal Requirements:

3. "Thank you for stating the following financial disclosure: 

"N/A"

Please include your amended statements within your cover letter; we will change the online submission form on your behalf."

**Comments to the Author**

1. Is the manuscript technically sound, and do the data support the conclusions?

Reviewer #1: Yes

Reviewer #2: Yes

2. Has the statistical analysis been performed appropriately and rigorously? 

Reviewer #1: Yes

Reviewer #2: Yes

3. Have the authors made all data underlying the findings in their manuscript fully available?

Reviewer #1: Yes

Reviewer #2: Yes

4. Is the manuscript presented in an intelligible fashion and written in standard English?

Reviewer #1: Yes

Reviewer #2: Yes

5. Review Comments to the Author

Reviewer #1: The manuscript is well presented, the topics innovative and well described. The tables and graphs are clear

and sufficient. Only a few aspects would need to be revised by the authors:

•It would be advisable to include the abbreviations in the abstract (lines 28-29).

•It is suggested that the authors include information on the epidemiology, staging and diagnosis, tumour location and mortality rate of OSC in the introduction. In this regard, the authors may also refer to the following work: de Nigris F, Rossiello R, Schiano C, Arra C, Williams-Ignarro S, Barbieri A, Lanza A, Balestrieri A, Giuliano MT, Ignarro LJ, Napoli C; Deletion of Yin Yang 1 Protein in Osteosarcoma Cells on Cell Invasion and CXCR4/Angiogenesis and Metastasis. Cancer Res 15 March 2008; 68 (6): 1797–1808. https://doi.org/10.1158/0008-5472.CAN-07-5582.

•It would be appropriate to clarify in the discussion the possible future developments of the project and how to proceed in order to solve the limitations of this study.

Reviewer #2: The study is aimed to predict the cancer-specific survival rate in patients with OSC using deep learning algorithms and classical Cox proportional hazard models to provide data to support individualized treatment of patients with OSC. Data derived from patients diagnosed with OSC from 2004 to 2017 and obtained from the Surveillance, Epidemiology, and End Results database. The study sample was then divided randomly into a training cohort and a validation cohort in the proportion of 7:3. The DeepSurv algorithm and the Cox proportional hazard model were chosen to

construct prognostic models for patients with OSC. A total of 3218 patients were randomized into training and validation groups (n = 2252 and 966, respectively). Both DeepSurv and Cox models had better efficacy in predicting cancer-specific survival (CSS) in OSC patients (C-index >0.74). In the validation of other metrics, DeepSurv did not have superiority over the Cox model in predicting survival in OSC patients.

I found the study of interest. The methodology is correct. Data are well presented and results support conclusions.

6. PLOS authors have the option to publish the peer review history of their article (what does this mean?). If published, this will include your full peer review and any attached files.

Reviewer #1: No

Reviewer #2: No

---

## [Author Response · Author response to Decision Letter 0]

10 May 2023

Reviewer #1 

•It would be advisable to include the abbreviations in the abstract (lines 28-29).

Response: Dear Prof. Thanks for your valuable comments. We have added the abbreviations of the relevant terms in the abstract. I would appreciate your review of the revised manuscript. (In red, the same below)

•It is suggested that the authors include information on the epidemiology, staging and diagnosis, tumour location and mortality rate of OSC in the introduction. In this regard, the authors may also refer to the following work: de Nigris F, Rossiello R, Schiano C, Arra C, Williams-Ignarro S, Barbieri A, Lanza A, Balestrieri A, Giuliano MT, Ignarro LJ, Napoli C; Deletion of Yin Yang 1 Protein in Osteosarcoma Cells on Cell Invasion and CXCR4/Angiogenesis and Metastasis. Cancer Res 15 March 2008; 68 (6): 1797–1808. https://doi.org/10.1158/0008-5472.CAN-07-5582

Response: Thank you very much for the excellent reference you recommended, from which we have learned a lot of professional knowledge. The content related to OSC epidemiology is described in the text. We request you to review it again (Line 42-44).

•It would be appropriate to clarify in the discussion the possible future developments of the project and how to proceed in order to solve the limitations of this study.

Response: We have discussed experimental designs that may address manuscript flaws in the future (Line 233-234). Please review it and feel free to contact us if there are any inappropriate points. We look forward to more guidance from you.

Reviewer #2 

The study is aimed to predict the cancer-specific survival rate in patients with OSC using deep learning algorithms and classical Cox proportional hazard models to provide data to support individualized treatment of patients with OSC. Data derived from patients diagnosed with OSC from 2004 to 2017 and obtained from the Surveillance, Epidemiology, and End Results database. The study sample was then divided randomly into a training cohort and a validation cohort in the proportion of 7:3. The DeepSurv algorithm and the Cox proportional hazard model were chosen to construct prognostic models for patients with OSC. A total of 3218 patients were randomized into training and validation groups (n = 2252 and 966, respectively). Both DeepSurv and Cox models had better efficacy in predicting cancer-specific survival (CSS) in OSC patients (C-index >0.74). In the validation of other metrics, DeepSurv did not have superiority over the Cox model in predicting survival in OSC patients.

I found the study of interest. The methodology is correct. Data are well presented and results support conclusions.

Response: Dear Professor, Thank you for your comments. If there is anything else in the manuscript that is not appropriate, please feel free to contact us. We look forward to more guidance from you. Thanks again.

---

## [Editor Report · Decision Letter 1]

25 May 2023

A deep learning algorithm with good prediction efficacy for cancer-specific survival in osteosarcoma: a retrospective study

PONE-D-23-11162R1

Dear Dr. Xia

We’re pleased to inform you that your manuscript has been judged scientifically suitable for publication and will be formally accepted for publication once it meets all outstanding technical requirements.

Kind regards,

Filomena de Nigris, Ph.D.

Academic Editor

PLOS ONE
---

## [Editor Report · Acceptance letter]

31 May 2023

PONE-D-23-11162R1 

A deep learning algorithm with good prediction efficacy for cancer-specific survival in osteosarcoma: a retrospective study 

Dear Dr. Xia:

I'm pleased to inform you that your manuscript has been deemed suitable for publication in PLOS ONE. Congratulations! Your manuscript is now with our production department. 

Kind regards, 

on behalf of

Prof. Filomena de Nigris 

Academic Editor

PLOS ONE